# An atlas of amyloid aggregation: the impact of substitutions, insertions, deletions and truncations on amyloid beta fibril nucleation

Mireia Seuma[1], Ben Lehner [2,3,4,5] ✉ & Benedetta Bolognesi [1] ✉

Multiplexed assays of variant effects (MAVEs) guide clinical variant interpretation and reveal disease mechanisms. To date, MAVEs have focussed on a single mutation type–amino acid (AA) substitutions–despite the diversity of coding variants that cause disease. Here we use Deep Indel Mutagenesis (DIM) to generate a comprehensive atlas of diverse variant effects for a disease protein, the amyloid beta (Aβ) peptide that aggregates in Alzheimer's disease (AD) and is mutated in familial AD (fAD). The atlas identifies known fAD mutations and reveals that many variants beyond substitutions accelerate Aβ aggregation and are likely to be pathogenic. Truncations, substitutions, insertions, single- and internal multi-AA deletions differ in their propensity to enhance or impair aggregation, but likely pathogenic variants from all classes are highly enriched in the polar N-terminal region of Aβ. This comparative atlas highlights the importance of including diverse mutation types in MAVEs and provides important mechanistic insights into amyloid nucleation.

Amyloid fibrils are the hallmarks of more than 50 human diseases, including Alzheimer's disease (AD), Parkinson's disease, frontotemporal dementia, amyotrophic lateral sclerosis, and systemic amyloidosis[1]. Mutations in the proteins that aggregate in the common forms of neurodegeneration also cause rare familial neurodegenerative diseases. For example, amyloid plaques of the amyloid beta (Aβ) peptide are a pathological hallmark of AD and specific dominant mutations in Aβ also cause familial Alzheimer's disease (fAD)[2,3].

The structures of many amyloid fibrils have now been determined[4], including those of Aβ fibrils extracted post-mortem from AD patient brains[5]. In these fibrils, the first part of the peptide remains unstructured (residues 1–9 in sporadic AD and 1–11 in fAD) while residues from 12 onwards arrange into a set of beta strands with an S-shaped fold[4]. Despite these high-resolution structures, the mechanism by which fibrils form in the first place–the nucleation reaction–is still poorly understood, even though this is the fundamental process that needs to be understood and targeted to prevent amyloid diseases[6,7]. Moreover, we have only a superficial

understanding of how specific mutations accelerate the process of amyloid nucleation to cause familial diseases. One of the hypotheses is that these mutations accelerate the formation of toxic Aβ oligomeric species, which are very difficult to capture and characterize by classic in vitro methods[8]. In Aβ, several of the known fAD mutations are at residues outside the structured amyloid core[9]. Amongst other consequences, this makes the clinical interpretation of genetic variants challenging, with the vast majority of mutations identified in aggregating proteins classified as variants of uncertain significance (VUSs)[10].

Multiplexed assays of variant effects (MAVEs)[11] use cell-based or in vitro selection assays to build comprehensive atlases of variant effects (AVEs)[12] to guide the clinical interpretation of VUSs[12]. This approach, which is also called deep mutational scanning (DMS), uses massively parallel DNA synthesis, selection, and deep sequencing to quantify the relative activities of variants in a functional assay[13]. Applied to disease genes, DMS can also reveal disease mechanisms and it can be used to genetically-validate the relevance of cellular and

[1]Institute for Bioengineering of Catalonia (IBEC), The Barcelona Institute of Science and Technology, Baldiri Reixac 10-12, 08028 Barcelona, Spain. [2]Centre for Genomic Regulation (CRG), The Barcelona Institute of Science and Technology, Doctor Aiguader 88, 08003 Barcelona, Spain. [3]Universitat Pompeu Fabra (UPF), Barcelona, Spain. [4]ICREA, Pg. Lluís Companys 23, Barcelona 08010, Spain. [5]Wellcome Sanger Institute, Wellcome Genome Campus, Hinxton, UK. ✉e-mail: ben.lehner@crg.eu; bbolognesi@ibecbarcelona.eu

in vitro disease models to human disease[14]. For example, we recently adopted a cell-based assay[15] which employs yeast to allow massively parallel quantification of variant effects on protein aggregation. Measuring the effects of single nucleotide changes in Aβ revealed that the assay both accurately quantifies the rate of amyloid fibril nucleation and that it identifies all of the dominant substitutions known to cause fAD[16].

To date, MAVE experiments[12] have focussed on a single type of mutation−amino acid (AA) substitutions−and have largely ignored additional forms of genetic variation. Insertions and deletions (indels), in particular, are an abundant and important class of genetic variation in protein-coding regions known to cause many human genetic diseases[17,18], with small indels (<21 bp) causing approximately 24% of Mendelian diseases[19,20]. Indels are a fundamentally different perturbation to a protein sequence to substitutions: whereas substitutions only alter AA side chains, indels are backbone mutations that change the length of the polypeptide chain and so may be expected to have more severe effects[21]. However, despite their importance, there has been very little systematic quantification of the effects of indels in proteins[22–24], particularly in disease genes, and many computational methods for predicting variant effects simply ignore them[25]. To our knowledge, a systematic comparison of the effects of AA substitutions, insertions, and deletions is lacking for any human disease gene.

Here we address this shortcoming in human genetics by providing a comprehensive comparison of the effects of substitutions, insertions, and deletions in a human disease gene.

The resulting AVE quantifies the effects of diverse sequence changes on the aggregation of Aβ and can be used to guide the clinical interpretation of different types of mutation in a human disease gene. It reveals that many mutations beyond substitutions accelerate the aggregation of Aβ and so are likely to be pathological. The atlas identifies the two deletions known to cause fAD, but reveals that they are only two of the many insertions and deletions that are likely to be pathological. The atlas also provides mechanistic insight into the process of amyloid nucleation, illustrating the power of deep indel mutagenesis (DIM) to illuminate sequence-to-activity relationships.

## Results
### Deep indel mutagenesis of Aβ
To quantify and contrast the effects of diverse genetic variants on the aggregation of the 42 AA form of Aβ (Aβ42), which is the most abundant component of amyloid plaques in AD, we performed deep indel mutagenesis (DIM) by synthesizing a library containing all possible single AA substitutions ($n = 798$), all possible single AA insertions ($n = 780$), all single AA deletions ($n = 37$), all internal multi-AA deletions ranging in length from 2–39 AA ($n = 731$), and all progressive truncations from the N-terminus, C-terminus or both, removing 2–39 AA ($n = 817$, Fig. 1a).

We quantified the effects of these different classes of variants in a cell-based selection assay where the aggregation of Aβ nucleates the aggregation of an endogenous protein, a process required for growth in selective conditions (Fig. 1a)[15,16]. After selection, the enrichment of each variant in the library was quantified by deep sequencing[16,26]. The resulting enrichment scores are reproducible between replicates (Supplementary Fig. 1a) and correlate well with previous measurements (R = 0.82, Supplementary Fig. 1b) as well as with the effects of variants quantified individually (R = 0.89, Supplementary Fig. 1c). Testing individual variants revealed that their growth rates were not confounded by differential toxicity (Supplementary Fig. 1f). We, and others, have also previously shown that expression of the Sup35N domain alone does not result in amyloid nucleation[15,16]. In addition, and as previously reported[16], the enrichment scores correlate linearly with the in vitro measured kinetic rate constants of Aβ amyloid fibril nucleation (Fig. 1b and

Supplementary Fig. 1d and 1e, Supplementary Data 1)[16,27,28], so we refer to them as "nucleation scores". More specifically, the nucleation scores correlate with the kinetic rate constants of secondary nucleation, the recognized rate-limiting microscopic mechanism in the Aβ42 aggregation reaction[29] (R = 0.73, Fig. 1b and Supplementary Fig. 1e).

### Contrasting the impact of substitutions, deletions, and insertions in a disease gene
The resulting dataset provides an opportunity to comprehensively compare the effects of different types of mutation−substitutions, insertions, deletions, and truncations−in a human disease gene. Focussing on single AA changes, the most frequent mutational effect is reduced aggregation, with 43% of substitutions, 44% of insertions, and 37% of deletions having lower nucleation scores (NS) than wild-type (WT) Aβ (false discovery rate, FDR = 0.1, NS− variants, Fig. 1c, e). The effects of multi-AA deletions are stronger, with 60% of internal multi-AA deletions and 97% of multi-AA truncations from one or both ends reducing nucleation (FDR = 0.1, Fig. 1c, e).

### Many variants beyond substitutions accelerate Aβ aggregation
Variants in Aβ identified in families with fAD accelerate Aβ aggregation, consistent with a gain-of-function mechanism[16,30]. Unlike computational methods to predict aggregation or variant effects, the experimental nucleation scores accurately classify fAD variants (area under the receiver operating characteristic curve, AUC = 0.88, Supplementary Fig. 1g). In total, there are 307 variants in our library (10%) that accelerate Aβ aggregation (FDR = 0.1, NS + variants): 108 substitutions, 77 insertions, 5 single AA deletions, 104 internal multi-AA deletions and 13 truncations (Fig. 1g, h and Supplementary Data 2). There are thus many variants beyond substitutions that accelerate the aggregation of Aβ.

### All types of variant that promote aggregation are enriched in the 1−28 N-terminal region
The primary sequence of Aβ contains two regions composed entirely of aliphatic residues and glycines (AA 17–21 and AA 29–42) (Fig. 1a). These stretches have in the past been identified as aggregation-prone regions of the peptide (APR1, APR2)[31,32]. The first 9 or 11 residues, predominantly polar or charged, are unstructured in Aβ fibrils obtained from sporadic and fAD brains, respectively[5]. The other two polar and charged stretches, residues 12–16 and 22–28, connect this first disordered region to APR1, and APR1 to APR2, respectively (Fig. 1a). We explored mutational impact in these five regions separately (Fig. 1d and Supplementary Fig. 1h).

For all classes of mutation, variants that reduce nucleation are strongly enriched in AA 29–42 (APR2): 60% of substitutions, 54% of insertions, 78% of single AA deletions, 85% of the internal multi-AA deletions, and all truncations that reduce nucleation occur in this hydrophobic region (FDR = 0.1; Fig. 1d, g and Supplementary Fig. 1i). Indeed for all mutation types, the majority of variants in this region impair nucleation: 76% of substitutions, 76% of insertions, all single AA deletions, 94% of internal multi-AA deletions and all truncations (Fig. 1f).

In contrast, variants that accelerate nucleation are strongly enriched in the N-terminal region of the peptide (AA 1−28), which includes the polar and charged stretches, as well as APR1 (Fig. 1d). The previously described APR1 and APR2 therefore display very different tolerance to mutations: APR1 resembles the rest of the 1−28 region, while most mutations in APR2 reduce nucleation (Fig. 1d). In total, 87% of variants that accelerate nucleation (267/307, FDR = 0.1, NS + variants) are located in the N-terminal region up to residue 28 (Fig. 1g). This contrasts to just 18% of variants in this region that reduce nucleation (Fig. 1g). This strong

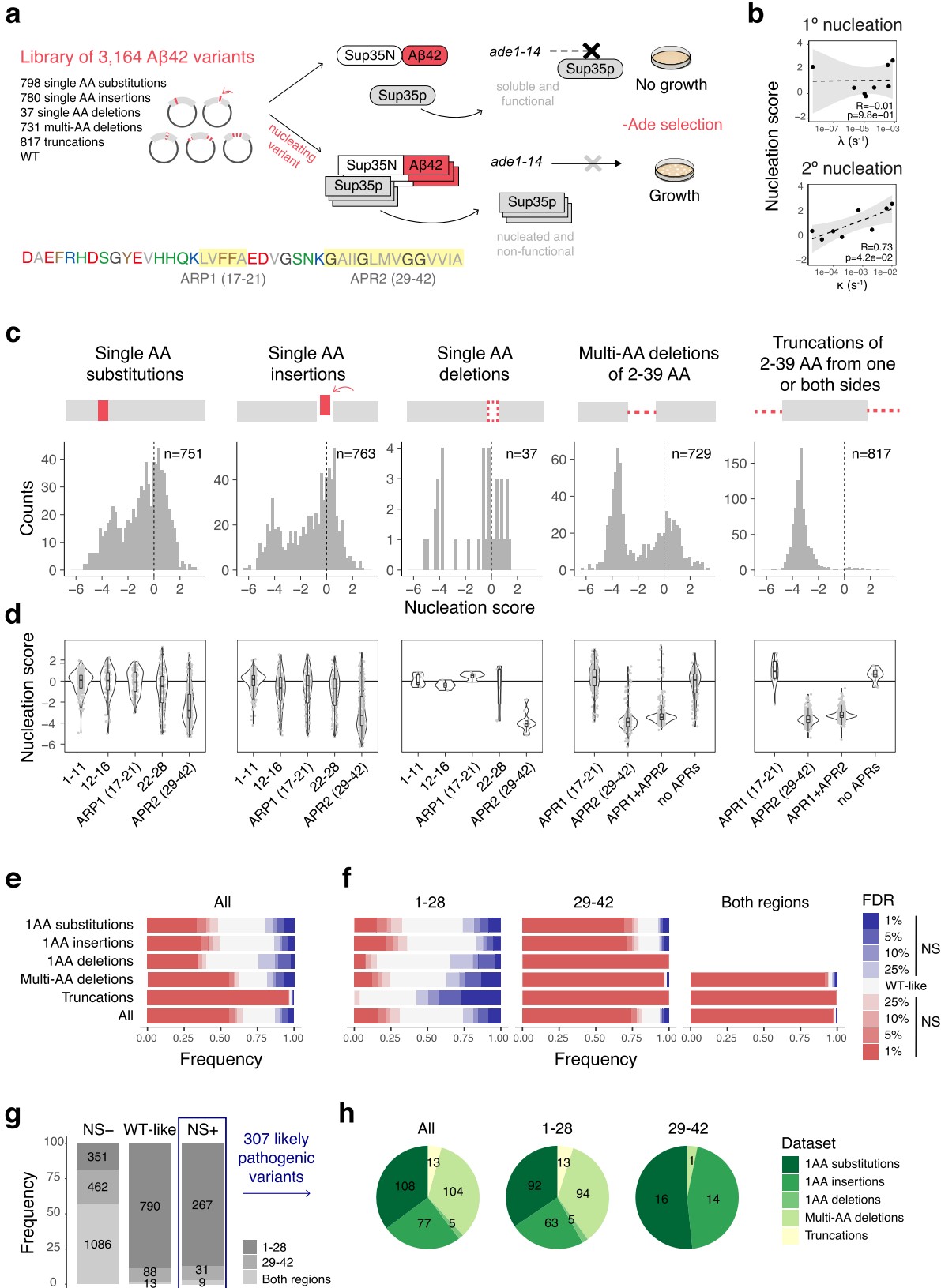

enrichment is true for all mutation types: 85% of substitutions, 82% of insertions, 90% of multi-AA deletions, all single AA deletions, and all truncations that accelerate nucleation occur in the N-terminal region (Fig. 1d, g, h). On the other hand, very few variants in APR2 (AA 29–42) increase nucleation: only 16 substitutions (6%) and 14 insertions (5%), while none of the single AA deletions do so. Similarly, no C-terminal truncations accelerate nucleation and only one internal multi-AA deletion in this region does so (Fig. 1f–h).

**Fig. 1 | Deep indel mutagenesis of Aβ. a** Aβ coding sequence colored by AA class (red: negative, blue: positive, green: polar, gray: aliphatic, brown: aromatic, dark gray: glycine. APRs are highlighted in yellow) and schematics of the in vivo selection assay. Aβ, fused to sup35N, seeds aggregation of sup35p causing a read-through of a premature stop codon in the *ade1* reporter gene allowing growth in medium lacking adenine. **b** Correlation of nucleation scores with in vitro primary (λ) and secondary (κ) nucleation rate constants from HDX-MS measurements[28,29]. Weighted Pearson correlation coefficients are indicated. Gray band indicates 95% confidence interval of the linear fits. **c** Distribution of nucleation scores for each class of mutations. Dashed lines indicate WT nucleation score (0). **d** Distributions of nucleation scores for mutations in different regions as reported in refs. 5, 31 ($n_{1\text{-}11\ subs} = 201$, $n_{12\text{-}16\ subs} = 86$, $n_{17\text{-}21\ subs} = 90$, $n_{22\text{-}28\ subs} = 124$, $n_{29\text{-}42\ subs} = 250$, $n_{1\text{-}11\ ins} = 205$, $n_{12\text{-}16\ ins} = 91$, $n_{17\text{-}21\ ins} = 94$, $n_{22\text{-}28\ ins} = 132$, $n_{29\text{-}42\ ins} = 241$, $n_{1\text{-}11\ s.del} = 11$, $n_{12\text{-}16\ s.del} = 4$, $n_{17\text{-}21\ s.del} = 4$, $n_{22\text{-}28\ s.del} = 7$, $n_{29\text{-}42\ s.del} = 11$, $n_{17\text{-}21\ multi\text{-}del} = 212$, $n_{29\text{-}42\ multi\text{-}del} = 150$, $n_{APR1+APR2\ multi\text{-}del} = 246$, $n_{no\ APR\ multi\text{-}del} = 121$, $n_{17\text{-}21\ trunc} = 12$, $n_{29\text{-}42\ truncl} = 356$, $n_{APR1+APR2\ trunc} = 434$, $n_{no\ APR\ trunc} = 15$ genotypes). Boxplots represent median values and the lower and upper hinges correspond to the 25th and 75th percentiles, respectively. Whiskers extend from the hinge to the largest value no further than 1.5x interquartile range. **e, f** Frequency of variants increasing or decreasing nucleation at different FDRs for the full peptide (**e**) and for peptide regions 1–28 and 29–42 (**f**). **g** Frequency and total counts of each mutation type for variants increasing (NS+), decreasing (NS−) or having no effect (WT-like) at FDR = 0.1. **h** Number and type of variants increasing nucleation (NS+) for each peptide region. Source data are provided as a Source data file.

## Mutation classes differ in their propensity to promote or prevent amyloid nucleation

The different classes of mutation do, however, vary in how likely they are to increase or decrease nucleation when they occur in the same region. The type of mutation most likely to accelerate nucleation is N-terminal truncations, i.e., truncations that start from AA 1–28, with 50% increasing nucleation and no N-terminal truncation reducing nucleation (FDR = 0.1, Fig. 1f). More internal multi-AA deletions in the N-terminus increase than decrease nucleation (28% vs. 19%), as do more single AA deletions (19% vs 11%). In contrast, single AA substitutions in this region are more likely to decrease (26%) than increase (18%) nucleation, as are insertions (30% decrease and 12% increase) (FDR = 0.1, Fig. 1f).

In summary, the DIM data reveals that there are many mutations beyond single AA substitutions that accelerate Aβ aggregation and so are potentially pathogenic (Supplementary Data 2). Moreover, they show that, for all mutation types, the vast majority of variants that accelerate nucleation are located in the N-terminal region of Aβ, up to residue 28. However, even here, the different classes of mutation have very different distributions of mutational effects: whereas single AA substitutions and insertions in the N-terminal region are more likely to decrease nucleation than increase it, the opposite is true for single AA deletions, internal multi-AA deletions, and N-terminal truncations: these mutation classes more often enhance nucleation than impair it, suggesting they are particularly likely to be pathogenic if they occur.

## AA preferences in the 1–28 N-terminal region: polar, positive, small, and P residues promote nucleation

Considering all positions, the effect of substituting in an AA is moderately correlated to the effect of inserting the same AA before or after the same position (R = 0.49 and R = 0.51, respectively, Fig. 2e). This relationship is, however, partly driven by the distinct impact of mutations before residue 28 compared to the C-terminal portion of the peptide. (Supplementary Fig. 2). Thus, although the consequences of insertions and substitutions are related, they are also clearly distinct, as is also revealed by comparing their effects at each individual residue (Supplementary Fig. 3a) and their average effects across all residues (Supplementary Fig. 4a).

Considering the 1–28 and 29–42 regions separately, the average effects of inserting or substituting in AAs in any positions are strongly related (R = 0.91 and R = 0.73, respectively, Fig. 2f and Supplementary Fig. 4b, c). Both substituting and inserting polar residues (especially, N,H,T,Q) into the N-terminal portion of Aβ frequently promotes aggregation, as does adding positively charged residues (K,R). Interestingly, substituting in or inserting G or A into the N-terminus also frequently increases nucleation as does adding a P, particularly between AA 9 and 23 (Fig. 2a, b, d and Supplementary Figs. 5a, b and 6–8). Since P residues are unlikely to be tolerated in the core of structured fibrils, their effect in promoting nucleation may be via changes in the ensemble of soluble Aβ[33], rather than due to changes in the fibril transition state. For example, adding P might impair the formation of a transient secondary structure that—in the WT ensemble—acts to prevent nucleation.

Overall, these enrichments for polar, positively charged, small and P residues are very different to the sequence preferences used by computational methods to predict protein aggregation[31,34–36], and these methods indeed perform very poorly for predicting the effects of mutations in the N-terminus of Aβ (Supplementary Fig. 9). This poor predictive performance may be due to the small and biased datasets used to train the existing computational predictors of protein aggregation and highlights the need to generate larger and more diverse experimental datasets to train the next generation of computational methods.

## AA preferences in the 1–28 N-terminal region: increased hydrophobicity and negatively charged residues reduce aggregation

Also inconsistent with the expectations of predictive methods, the substitutions and insertions in the N-terminus that most often reduce aggregation are additions of hydrophobic (W,L,F,M,I,Y,V) and negatively charged (D,E) AAs (Fig. 2a, b, d and Supplementary Fig. 5a, b). Consistent with this, individually deleting negatively charged residues and L17 often increases nucleation (Fig. 2c), as does substituting away from these same AAs (Supplementary Fig. 5c).

However, there are many exceptions to these general trends, highlighting the importance of generating the full mutational matrix. For example, W insertions and substitutions to W mostly promote aggregation in residues 1–12 but nearly always impair aggregation at positions 13–28 (Fig. 2a, b and Supplementary Fig. 5a, b). In addition, many substitutions to V and I in positions 13–20 strongly increase aggregation, as do many hydrophobic insertions after residue 13 between two histidines. The specific positioning of a hydrophobic residue (V) between two histidines has been proposed as a zinc-binding motif that promotes association of strands and nucleation of a designed amyloid sequence[37]. At particular positions the impact of substitutions or insertions can also be quite distinct: for example substitutions of F19 and F20 rarely increase nucleation and only for mutations to hydrophobic AA, while many insertions between F19 and F20 increase nucleation, especially of charged and polar residues (Fig. 2a, b, and Supplementary Fig. 5a, b). At other residues the preferences are more similar: for example both substitutions to and insertions of G at residues 22 and 23 increase nucleation resulting in some of the fastest nucleating variants in the library (Fig. 2a, b and Supplementary Figs. 5a, b and 10), suggesting that increasing flexibility or reducing side chain volume in this region favors nucleation.

Thus, although simple rules can predict mutational effects to some extent, the comprehensive Aβ data suggests full experimental datasets and new computational methods will be required for the clinical interpretation of variants in aggregating proteins.

In summary, the role of the N-terminal two-thirds (AA 1–28) of Aβ in promoting and preventing amyloid nucleation must be very different to that of the C-terminus that forms the hydrophobic core of Aβ fibrils (AA 29–42). To our knowledge, no existing mechanistic models can satisfactorily account for mutational effects in this region[38,39]. In

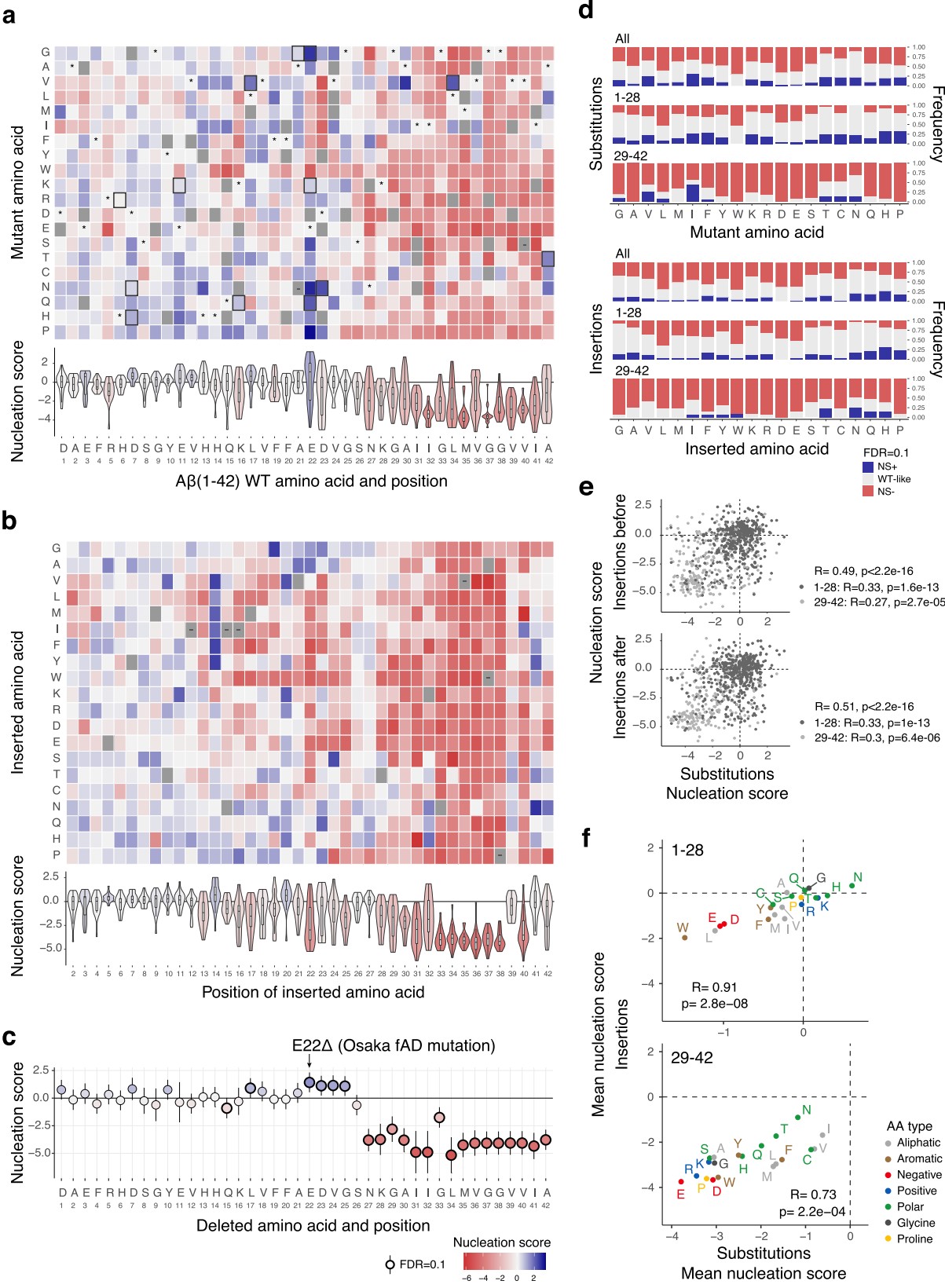

mature Aβ fibrils derived from patients[5] and formed in vitro[5,40–44], part of the N-terminus remains unstructured (see below). Changes in aggregation caused by mutations in this region could be due to their effects on the ensemble of soluble Aβ. Alternatively, the N-terminus could participate directly in the nucleation reaction, establishing interactions in the nucleation transition state.

**The Osaka mutation (E22Δ) is the fastest nucleating single AA deletion**

To date, only one single AA deletion has been reported in families with fAD: deletion of residue E22, named the Osaka mutation after the city in which it was first identified[45]. Strikingly, our data shows that the Osaka mutation is the single AA deletion that most enhances the

**Fig. 2 | Single AA variant atlases. a** Heatmap of nucleation scores for single AA substitutions. The WT AA and position are indicated in the x-axis and the mutant AA is indicated in the y-axis. Variants not present are indicated in gray, synonymous mutants with '*' and fAD mutants with a black box. Non-nucleating variants (with no NS, see "Methods") are indicated with '-'. The distribution of nucleation scores for each position is summarized in the violin plots below the heatmap. **b** Heatmap of nucleation scores for single AA insertions at each position. **c** Effect of single AA deletions ($n = 3$ biological replicates). The horizontal line indicates the WT nucleation score (0). Vertical error bars indicate 95% confidence interval of the mean. **d** Frequency of AA increasing or decreasing nucleation (FDR = 0.1) upon substitutions (top) or insertion (bottom) for each peptide region. **e** Correlation of nucleation scores for substitutions to each AA at each position and insertions of the same AA before (top, $p = 2.3e{-}45$) or after (bottom, $p = 2e{-}49$) that position. Color indicates peptide region (1–28 and 29–42). **f** Correlation of average nucleation scores for each AA, for insertions and substitutions at the N-terminus (top) and at the C-terminus (bottom). Color indicates AA type. Pearson correlation coefficients are indicated in (**e**) and (**f**).

nucleation of Aβ (Fig. 2c). However, an additional 4 single AA deletions promote nucleation (FDR = 0.1), suggesting that they may also be pathogenic. All of these consist of deletions before residue 25 (Fig. 2c).

### The Uppsala mutation (Δ19–24) lies in a hotspot of internal multi-AA deletions that promote Aβ nucleation

After we generated this dataset, the first internal multi-AA deletion in Aβ that causes fAD was reported[46]. This deletion, referred to as the Uppsala mutation, removes AA 19–24. Strikingly, the Uppsala mutation is in the centre of a large hotspot of multi-AA deletions that promote nucleation in our dataset (Fig. 3a–d). In total, 35 different multi-AA deletions removing some or all of residues 17–27 increase the nucleation of Aβ (FDR = 0.1, Fig. 3d and Supplementary Fig. 11a), as do a total of 104 internal multi-AA variants. This suggests that there are potentially many more pathogenic deletions that remain to be discovered that remove residues in this central hotspot region, as well as additional pathogenic deletions throughout the N-terminus (Supplementary Data 2).

The multi-AA deletion hotspot is centered on the negatively charged residues E22 and D23 (Fig. 3a–d and Supplementary Fig. 11a). Many substitutions at these two positions also accelerate nucleation (Fig. 2a) as does the individual deletion at position E22 (Osaka mutation, Fig. 2c). However, not all internal multi-AA deletions that remove E22 or D23 increase aggregation, with deletions starting from positions 4, 5, 12, and 13 that remove E22 or D23 failing to accelerate nucleation (Fig. 3a). In these cases, a negatively charged residue is relocated to the immediate proximity—one or two residues away—of the aliphatic core (AA 29–42, Fig. 3a and Supplementary Fig. 11b), where they likely compensate for the loss of negative charge.

The importance of charge in mediating the effects of multi-AA deletions is also suggested by a cluster of deletions in the first 15 residues of Aβ that accelerate nucleation (Fig. 3a and Supplementary Fig. 11a). This region contains four of the negatively charged residues in Aβ (D1, E3, D7, and E11) with many substitutions of these residues also accelerating aggregation (Fig. 2a). The matrix of internal multi-AA deletions further reveals that deletions that remove D1 have higher NSs than those that keep it; the same is true for D7 (Fig. 3a and Supplementary Fig. 11a). This segment is unstructured in nearly all mature Aβ fibril polymorphs[40–44], including those in AD brains[5] (see below), yet diverse types of mutation in this region strongly increase aggregation.

### Mutations in the aliphatic region 29–42 that accelerate aggregation

The vast majority of mutations of any type within APR2, i.e., the aliphatic C-terminus (AA 29–42) of Aβ, strongly disrupt nucleation (Fig. 1d, f, g). Indeed all insertions in the 33–38 stretch disrupt nucleation, suggesting that this may constitute the inner core of the nucleation transition state (Fig. 2b and Supplementary Fig. 5b).

However, there are some variants in the C-terminus that increase nucleation: 16 substitutions, 14 insertions, one internal multi-AA deletion within this region, and nine multi-AA deletions that involve C-terminal residues (FDR = 0.1, Fig. 1g, h). The substitutions in the C-terminus that accelerate nucleation are enriched at A30 and A42. At position 42, mutations to L promote nucleation, as do changes to C, T, and N (Fig. 2a and Supplementary Fig. 5a). Among these, only A42T is a

known fAD variant (Supplementary Data 3). At position 34 and 36, 4 substitutions to alternative hydrophobic AAs promote nucleation, suggesting that the L and V side chains may not be optimal in the nucleation transition state. L34V is also a known fAD variant (Supplementary Data 3). Insertions that promote nucleation are also enriched at specific positions. Polar insertions at position 32, flanking G33, may favor a turn, and polar, aromatic and hydrophobic insertions at positions 39, 41 and 42 (Fig. 2b and Supplementary Fig. 5b) may be more easily accommodated by minor structural rearrangements.

The only deletion within the aliphatic core 29–42 that accelerates nucleation is the removal of G33 and L34, although the individual deletion of each of these residues disrupts nucleation (Fig. 3a and Supplementary Fig. 11a). It is possible that adjustments in the core can accommodate removal of these two residues by the formation of a similar structural polymorph. Finally, nine internal multi-AA deletions that bridge the 1–28 and 29–42 regions increase nucleation (FDR = 0.1, Fig. 3a, e and Supplementary Fig. 11a). These deletions remove aliphatic core residues but replace them with a similar number of aliphatic residues from APR1 (Fig. 3a, e). It is likely that these internal multi-AA deletions are therefore creating alternative aliphatic cores that nucleate to form the same or similar structural polymorphs as full-length Aβ. These alternative cores that increase nucleation have a specific range of core lengths, with the hydrophobic stretch spanning from 13 to 16AA, very similar to the length of the 29–42 region in the WT peptide (Fig. 3e, f and Supplementary Fig. 12a).

### Positive charge promotes the nucleation of a minimal Aβ core

The DIM dataset shows that progressively removing AAs from the N-terminus of Aβ generates many peptides that aggregate faster than the full 42AA isoform, with 13/27 N-terminal truncations promoting nucleation (Fig. 4a, b and Supplementary Fig. 13). Such N-terminally truncated fragments of Aβ have been detected in AD patients[47,48] (Supplementary Data 4). Our data suggests that environmental triggers, infections or genomic alterations that increase their production are likely to accelerate Aβ aggregation and so may be causally important in familial and sporadic AD.

In contrast, all N-terminal truncations that remove at least one residue of the aliphatic core (AA 29–42) very strongly reduce aggregation, further highlighting the critical requirement for this region in nucleation (Fig. 4a and Supplementary Fig. 13). Strikingly, however, the aliphatic core alone nucleates very slowly (FDR = 0.1, Fig. 4a). The addition of residue 28 to this minimal core dramatically accelerates nucleation, with the 15AA peptide consisting of residues 28–42 actually being the fastest nucleating N-terminally truncated form of Aβ (Fig. 4a, b). This minimal Aβ core nucleates faster than full-length Aβ (Fig. 4a) and is too short to form the S-shaped amyloid fibrils polymorph observed in AD plaques[5] and so likely adopts a smaller C-shaped polymorph with two main strands facing each other. The rapid nucleation of this 15AA peptide is particularly striking given the observation that all other multi-AA deletions of more than 23AA prevent nucleation (Fig. 3a).

Residue 28 is a lysine and many of the other faster nucleating N-terminally truncated peptides also have positively charged residues at or close to their N-termini (Fig. 4b). Moreover, internal multi-AA deletions that remove K28 but that still nucleate often have a positively

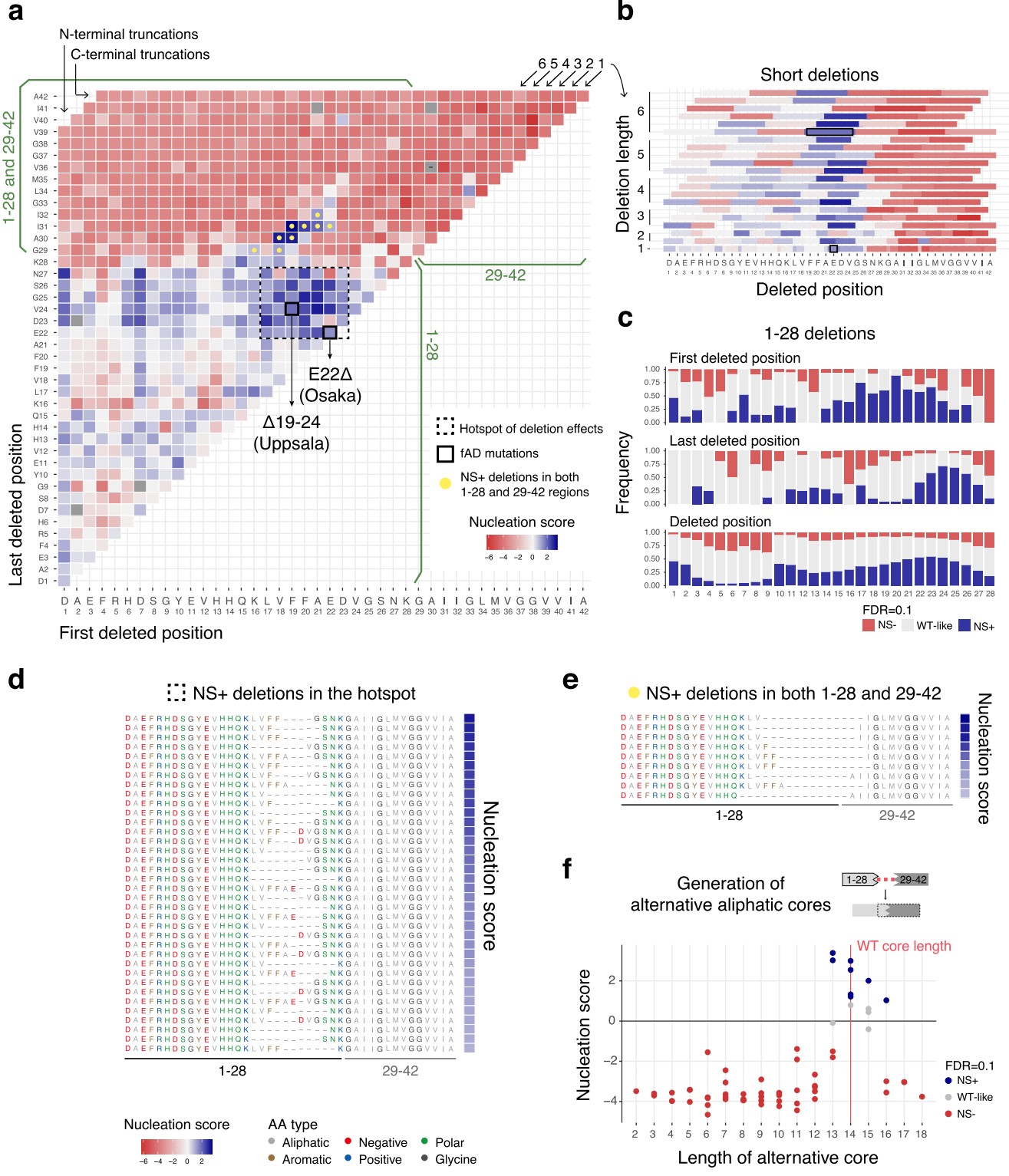

charged residue at the N-terminus of the 29–42 core (Supplementary Figs. 11c and 12b). We therefore tested the hypothesis that it is the addition of a positively charged residue that accelerates nucleation of the minimal aliphatic core of Aβ. Adding the positively charged residues K or R to the N-terminus of AA 29–42 strongly accelerated nucleation (Fig. 4c). In contrast, adding the negatively charged residues D or E did not (Fig. 4c). The addition of a single positively charged residue is therefore sufficient to dramatically accelerate the aggregation of the aliphatic core of Aβ. It is possible that positive residues, but not negative ones, at position 28, engage in a salt bridge with the

carboxyl group at the C-terminus of the peptide to promote nucleation[49].

## Residue 42 is required for fast nucleation

In contrast to the effects of N-terminal truncations, removing even a single AA from the C-terminus of Aβ strongly reduces nucleation (Fig. 4a and Supplementary Fig. 13). That A42 plays an important role in the nucleation of Aβ is consistent with previous reports that Aβ42 aggregates faster than Aβ40[6]. However, position 42 does not need to be an A: multiple substitutions and multiple insertions before position

**Fig. 3 | Internal multi-AA deletion atlas. a** Matrix of nucleation scores for deletions. The dashed-line black square depicts the hotspot of deletion effects (consecutive deleted positions where NS + frequency > ½ max(NS + frequency, i.e., deletions starting at positions 17–23 and ending at positions 22–27) and the yellow dots indicate deletions removing residues in both the N-terminus (AA 1–28) and C-terminus (AA 29–42) that increase NS (see (**e**)). Variants not present are represented in gray and non-nucleating variants (with no NS, see "Methods") are indicated with '-'. **b** Effect on nucleation of deletions of 1–6AA length. The WT AA and position are indicated in the x-axis. The black squares indicate fAD variants: Osaka (E22Δ) and Uppsala (Δ19–24). Color code as in (**a**). **c** Frequency of variants increasing nucleation (NS+), decreasing nucleation (NS−) or with no difference from WT (WT-like) at FDR=0.1, for sequences with a specific first deleted position

(i.e., each column in the matrix), last deleted position (i.e., each row in the matrix) or missing a specific residue, at the N-terminus (AA 1–28). **d** AA sequence for variants inside the hotspot of deletion effects with significantly increased NS (FDR = 0.1). **e** AA sequence for variants with internal multi-AA deletions removing residues from both the 1–28 and 29–42 regions, with significantly increased NS (FDR = 0.1). AA colored by AA class. Color code as in (**d**). **f** Nucleation scores of variants with putative alternative aliphatic cores of different lengths. The horizontal line indicates the WT nucleation score (0). Vertical red line indicates WT core length (14AA). Variants displaying alternative cores were defined as those internal multi-AA deletions removing residues in both the N and C-terminal regions that replace part of the 29–42 residues with exclusively aliphatic residues (n = 64).

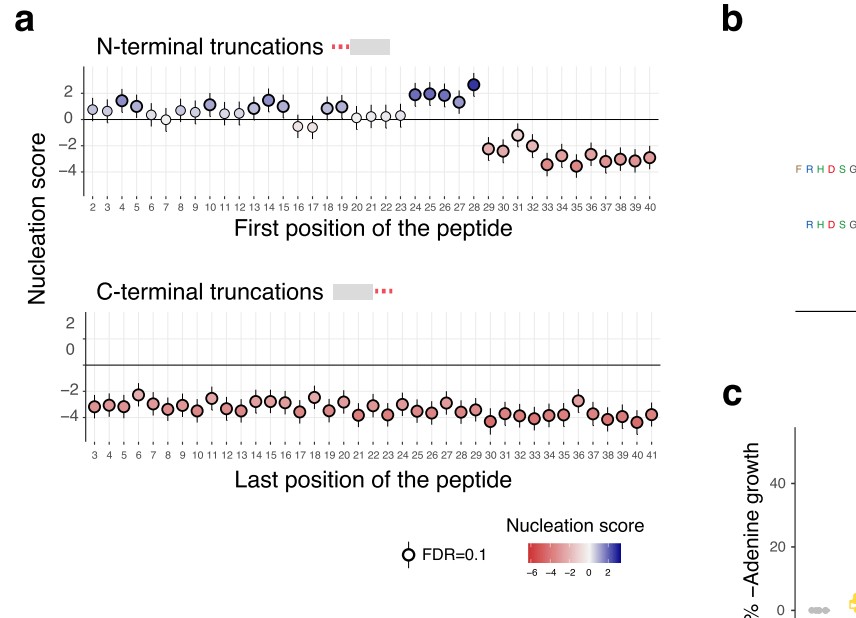

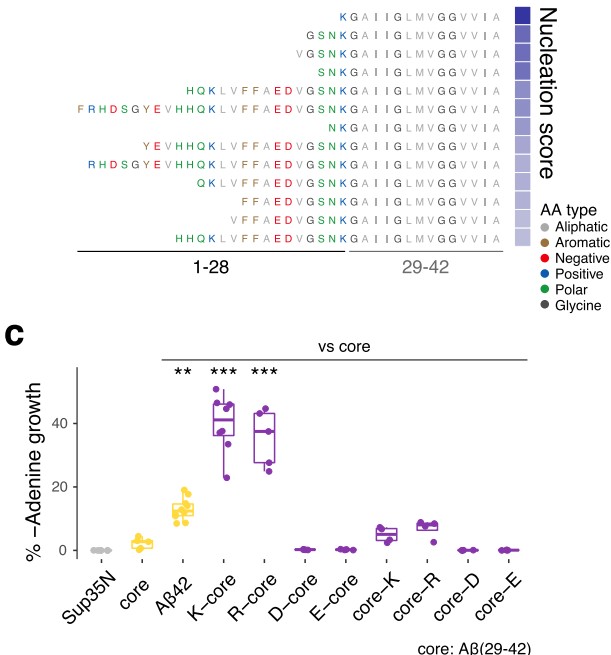

**Fig. 4 | Positive charge accelerates nucleation of a minimal Aβ core. a** Effect of N-terminal (top) and C-terminal (bottom) truncations on nucleation (n = 3 biological replicates). Vertical error bars indicate 95% confidence interval of the mean NS. **b** AA sequences of N-terminal truncations that increase nucleation at FDR = 0.1. AA are colored by class. **c** Effect of adding a positively or negatively charged residue at the N or C-terminus of the Aβ core (AA 29–42). Nucleation quantified as percentage of colonies in medium lacking adenine vs. medium containing adenine

(n > =4 biological replicates/variant). Boxplots represent median values and the lower and upper hinges correspond to the 25th and 75th percentiles, respectively. Whiskers extend from the hinge to the largest value no further than 1.5*inter-quartile range. One-way ANOVA with Dunnett's multiple comparisons test ($p_{Aβ42}$ = 1.16e−3, $p_{K-core}$ < 2.2e−16, $p_{R-core}$ = 6.6e−16). *$p$ < 0.05, **$p$ < 0.01, *** $p$ < 0.001. Source data are provided as a Source data file.

42 (Fig. 2a, b and Supplementary Fig. 5a, b) either do not disrupt nucleation or actually accelerate it. This suggests that the requirement for position 42 may therefore primarily be a steric one, for example to position a free carboxyl terminus in the nucleation transition state[5,49].

## Discussion

We have presented here a systematic comparison of the effects of substitutions, insertions, and deletions in a human disease gene. The resulting dataset shows that the consequences of AA insertions, deletions, and truncations are not trivial to predict from the effects of substitutions, highlighting the importance of including DIM when constructing an atlas of variant effects (AVE)[12] for the interpretation of clinical genetic variants.

The dataset provides a comprehensive AVE for Aβ aggregation that can be used to guide the future clinical interpretation of variants as they are discovered. The atlas reveals that many variants beyond substitutions accelerate the aggregation of Aβ and so are likely to be pathogenic. The identification of 307 variants that accelerate

aggregation (Supplementary Fig. 10 and Supplementary Data 2) in this very short 42AA peptide highlights the potentially enormous diversity of disease-causing variants in the human genome. For example, the Aβ AVE reveals that the Uppsala mutation (Δ19–24) is just one of many internal multi-AA deletions in a central hotspot region of Aβ that accelerate aggregation; these additional deletions are also likely to be pathogenic, as are multiple additional single AA deletions in the N-terminal region and many truncations of the peptide up to residue 28.

The substitutions, insertions, deletions, and truncations that accelerate nucleation are all strongly enriched in the first 28 residues of Aβ (Fig. 5 and Supplementary Fig. 14). The different classes differ, however, in their distributions of mutational effects, with substitutions and insertions in the N-terminus more likely to impair rather than enhance aggregation but single and multi-AA deletions more likely to enhance rather than impair it. N-terminal truncations of Aβ are particularly likely to accelerate nucleation, raising the intriguing possibility that increased production of N-terminally truncated forms of Aβ

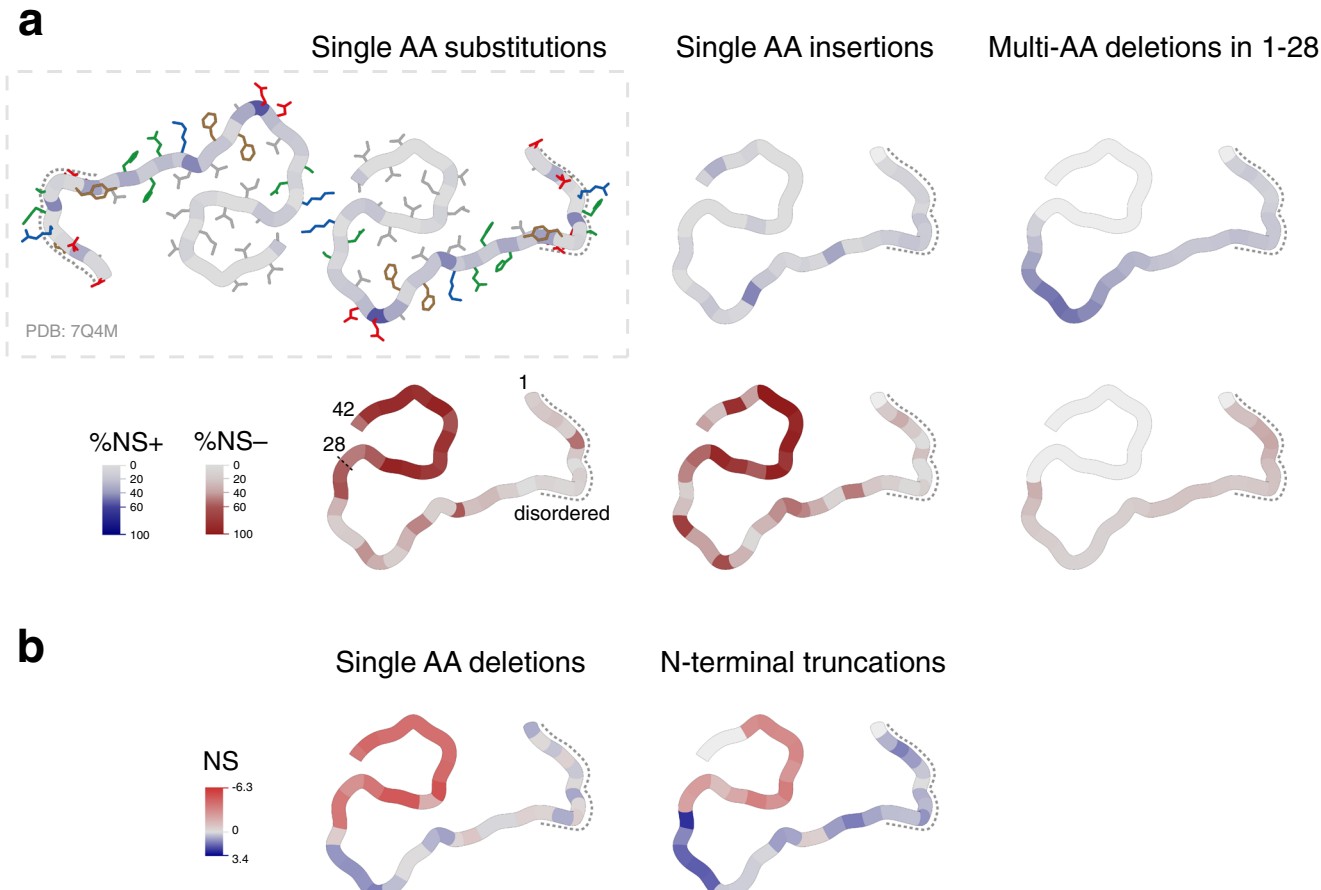

**Fig. 5 | Mutational effects visualized on fAD Aβ fibril structure.** In fibrils extracted from the brains of fAD patients, Aβ42 adopts an S-shaped structure at the C-terminus with an N-terminal arm linking to an unstructured region (AA 1–11) indicated by the dashed line (PDB: 7Q4M[5]). **a** Single AA substitutions, single AA insertions, and multi-AA deletions in AA 1–28: color intensity indicates the percentage of NS+ (blue) or NS− (red) mutations at each position or losing each position (for multi-AA deletions) (FDR = 0.1). Side chain colored by AA class (red: negative, blue: positive, green: polar, gray: aliphatic, brown: aromatic) **b** Single AA deletions and N-terminal truncations: color intensity depicts the nucleation score of each single AA deletion or of the N-terminal truncation starting at that position. White depicts positions that are not mutated in each dataset.

triggered by environmental exposures, pathogens or genetics might be an important cause of familial and sporadic AD[50].

The DIM dataset also provides substantial mechanistic insight into amyloid nucleation. The vast majority of mutations of any type in the aliphatic C-terminus of Aβ strongly reduce aggregation, which is consistent with this region forming the core of all known mature Aβ fibril polymorphs[5,40–44]. The exquisite sensitivity of this region to mutation suggests very strong structural constraints for amyloid nucleation. Only a few specific substitutions and insertions are tolerated, with these concentrated in residues at the end of the peptide which may more easily accommodate different side chains. In addition, the only internal multi-AA deletions that still nucleate despite removing residues from the C-terminal core are those that replace the missing residues with a similar number of aliphatic AA from a more N-terminal region of the peptide. The 14AA aliphatic core of Aβ, however, nucleates very poorly unless a positively charged residue is added at the N-terminus. We speculate that this charge may help solubilize this very hydrophobic peptide, prevent an alternative off-pathway non-amyloid aggregation process or participate directly in the nucleation transition state, for example by the formation of a salt bridge with the carboxy-terminus[49]. However, there is no correlation between an experimental measure of sequence solubility[51] and nucleation scores (Supplementary Fig. 9a).

In contrast, mutations in the N-terminus have much more diverse effects, with 267 variants in this region accelerating aggregation. This

holds for variants that involve the polar and charged residues which make up most of this region, but also for those with mutations in the aliphatic 17–21 stretch (APR1). Much of the N-terminus remains unstructured in mature Aβ fibrils, including those in AD amyloid plaques (Fig. 5 and Supplementary Fig. 14)[5,40–44]. Mutational effects in this region are not predicted by existing computational methods and they are not obviously interpretable using current mechanistic models of amyloid nucleation: polar, small, and positively charged residues as well as P tend to increase nucleation whereas hydrophobic and negatively charged residues tend to decrease it. However, these preferences can also be quite different at individual sites and in subregions of the N-terminus (Fig. 2a), further highlighting the importance of generating complete datasets for the interpretation of clinical variants.

We speculate that mutations in the polar N-terminus of Aβ alter the rate of nucleation because of effects on the ensemble of soluble Aβ or because they alter the propensity of the C-terminus to transition to a beta structure. The very strong enrichment of disease-causing and aggregation-promoting mutations in the N-terminus of Aβ makes understanding how polar extensions promote and prevent amyloid aggregation one of the highest priority goals in AD and amyloid research.

Finally, we envisage that the deep indel mutagenesis approach that we have applied here can be used to generate reference variant effect atlases and mechanistic insights for many other amyloid-

forming sequences, including those with a very different primary sequence to Aβ42.

## Methods

### Library design

The designed library contains a total of 3164 unique Aβ42 variants, with all single AA substitutions at each position ($n = 798$), all single AA insertions at all positions ($n = 780$), all deletions ranging from 1 to 39 AA in size in all positions ($n = 768$), sequences truncated from either one or both ends of the peptide with a minimum peptide length of 3 AA and maximum peptide length of 40 AA ($n = 817$), and the Aβ42 WT sequence ($n = 1$).

### Plasmid Library construction

The synthetic library was synthesized by Twist Bioscience and consisted of an Aβ42 variant region of 9 nt to 129 nt, flanked by 25 nt upstream and 21 nt downstream constant regions. 10 ng of the library were amplified by PCR (Q5 high-fidelity DNA polymerase, NEB) for 12 cycles with primers annealing to the constant regions (primers MS_01-02, Supplementary Data 5), according to the manufacturer's protocol. The product was then purified by column purification (MinElute PCR Purification Kit, Qiagen). In parallel, the $P_{CUP1}$-Sup35N-Aβ42 plasmid was linearized by PCR (Q5 high-fidelity DNA polymerase, NEB) with primers that remove the WT Aβ42 sequence (primers MS_03-04, Supplementary Data 5). The product was purified from a 1% agarose gel (QIAquick Gel Extraction Kit, Qiagen).

The library was then ligated into 100 ng of the linearized plasmid in a 5:1 (insert:vector) ratio by a Gibson approach with 3 h of incubation followed by dialysis for 45 min on a membrane filter (MF-Millipore 0.025 μm membrane, Merck). The product was transformed into 10-beta Electrocompetent *E. coli* (NEB), by electroporation with 2.0 kV, 200 Ω, 25 μF (BioRad GenePulser machine). Cells were recovered in SOC medium for 30 min and grown overnight in 30 ml of LB ampicillin medium. A small amount of cells were also plated in LB ampicillin plates to assess transformation efficiency. A total of 50,000 transformants were estimated, meaning that each variant in the library is represented >15 times. 5 ml of overnight culture were harvested to purify the Aβ42 library with a mini prep (QIAprep Miniprep Kit, Qiagen).

### Yeast transformation

*Saccharomyces cerevisiae* [psi-pin-] (MATa ade1–14 his3 leu2-3,112 lys2 trp1 ura3–52) strain provided by the Chernoff lab was used in all experiments in this study.

Yeast cells were transformed with the Aβ42 plasmid library in three biological replicates. An individual colony was grown overnight in 25 ml YPDA medium at 30 °C and 4 g. Cells were diluted in 150 ml to OD600 = 0.25 and grown for 4–5 h. When cells reached the exponential phase (OD-0.7–0.8), they were splitted in 10 transformation tubes of 15 ml each. Each tube was treated as follows: cells were harvested at $400 \times g$ for 5 min, washed with milliQ, and resuspended in 1 ml YTB (100 mM LiOAc, 10 mM Tris pH 8.0, 1 mM EDTA). They were harvested again and resuspended in 72 μl YTB. 100 ng of plasmid library were added to the cells, together with 8 μl of salmon sperm DNA (UltraPure, Thermo Scientific) previously boiled, 60 μl of dimethyl sulfoxide (Merck) and 500 μl of YTB-PEG (100 mM LiOAc, 10 mM Tris pH 8.0, 1 mM EDTA, 40% PEG 3350). Heat-shock was performed at 42 °C for 14 min in a thermo block. Finally, cells were harvested and resuspended in 300 ml plasmid selection medium (-URA, 20% glucose), pooling together the 10 transformation tubes and allowing them to grow for 50 h at 30 °C and 4 g. A small amount of cells were also plated in plasmid selection medium to assess transformation efficiency. A total of 118,125, 152,000, and 139,500 transformants were estimated for each biological replicate respectively, meaning that each variant in the library is represented >37 times.

After 50 h, cells were diluted in 25 ml plasmid selection medium to OD = 0.02 and grown exponentially for 15 h. Finally, the culture was harvested and stored at −80 °C in 25 % glycerol.

### Selection experiments

In vivo selection assays were performed in five technical replicates for each biological replicate. For each technical replicate, cells were thawed from −80 °C in 20 ml plasmid selection medium at OD = 0.05 and grown until exponential for 15 h. At this stage, cells were harvested and resuspended in 20 ml protein induction medium (-URA, 20% glucose, 100 μM Cu$_2$SO4) at OD = 0.05. After 24 h the 4x 5 ml input pellets were collected and 1 million cells/replicate were plated on -ADE-URA selection medium in 145-cm² plates (Nunc, Thermo Scientific). Plates were incubated at 30 °C for 7 days inside an incubator. Finally, colonies were scraped off the plates with PBS 1x and harvested by centrifugation to collect the output pellets. Both input and output pellets were stored at −20 °C for later DNA extraction.

### DNA extraction and sequencing library preparation

One input and one output pellets for each technical and biological replicate ($2 \times 5 \times 3$ samples) were resuspended in 0.5 ml extraction buffer (2% Triton-X, 1% SDS, 100 mM NaCl, 10 mM Tris-HCl pH 8, 1 mM EDTA pH 8). They were then frozen for 10 min in an ethanol-dry ice bath and heated for 10 min at 62 °C. This cycle was repeated twice. 0.5 ml of phenol:chloroform:isoamyl (25:24:1 mixture, Thermo Scientific) was added together with glass beads (Sigma). Samples were vortexed for 10 min and centrifuged for 30 min at $2000 \times g$. The aqueous phase was then transferred to a new tube, and mixed again with phenol:chloroform:isoamyl, vortexed and centrifuged for 45 min at $2000 \times g$. Next, the aqueous phase was transferred to another tube with 1:10 V 3 M NaOAc and 2.2 V cold ethanol 96% for DNA precipitation. After 30 min at −20 °C, samples were centrifuged and pellets were dried overnight. The following day, pellets were resuspended in 0.3 ml TE 1X buffer and treated with 10 μl RNAse A (Thermo Scientific) for 30 min at 37 °C. DNA was finally purified using 10 μl of silica beads (QIAEX II Gel Extraction Kit, Qiagen) and eluted in 30 μl elution buffer. Plasmid concentrations were measured by quantitative PCR with SYBR green (Merck) and primers annealing to the origin of replication site of the $P_{CUP1}$-Sup35N-Aβ42 plasmid at 58 °C for 40 cycles (primers MS_05-06, Supplementary Data 5).

The library for high-throughput sequencing was prepared in a two-step PCR (Q5 high-fidelity DNA polymerase, NEB). In PCR1, 50 million of molecules were amplified for 15 cycles with frame-shifted primers with homology to Illumina sequencing primers (primers MS_07–20, Supplementary Data 5). The products were purified with ExoSAP treatment (Affymetrix) and by column purification (MinElute PCR Purification Kit, Qiagen). They were then amplified for 10 cycles in PCR2 with Illumina-indexed primers (primers MS_21–37, Supplementary Data 5). The six samples of each technical replicate were pooled together equimolarly and the final product was purified from a 2% agarose gel with 20 μl silica beads (QIAEX II Gel Extraction Kit, Qiagen).

The library was sent for 125 bp paired-end sequencing in an Illumina HiSeq2500 sequencer at the CRG Genomics core facility. In total, >426 million paired-end reads were obtained, which is between 7 and 20 million per sample (i.e., input or output for a specific technical and biological replicate), representing >2200x read coverage for each designed variant in the library.

### Individual variant testing

Selected Aβ42 variants for individual testing were obtained by PCR linearisation (Q5 high-fidelity DNA polymerase, NEB) with mutagenic primers (primers MS_38–63, Supplementary Data 5). PCR products were treated with Dpn1 overnight and transformed in DH5α competent E.coli. Plasmids were purified by mini prep (QIAprep Miniprep Kit, Qiagen) and transformed into yeast cells using one transformation

tube of the transformation protocol described above. All constructions were verified by Sanger sequencing.

For %-adenine growth testing, yeast cells expressing individual variants were grown overnight in plasmid selection medium (-URA 20% glucose). They were then diluted to OD 0.05 in protein induction medium (-URA 20% glucose 100 μM Cu₂SO₄) and grown for 24 h. Cells were plated on -URA (control) and -ADE-URA (selection) plates in three independent replicates, and allowed to grow for 7 days at 30 °C. Adenine growth was calculated as the percentage of colonies in -ADE-URA relative to colonies in -URA.

For individual growth rate measurements, yeast cells expressing individual variants were grown overnight in plasmid selection medium (-URA 20% glucose) and diluted to OD 0.2 until exponential. They were then diluted again to OD 0.05 in non-inducing (-URA 20% glucose) and inducing (-URA 20% glucose 100 μM Cu₂SO₄) protein expression mediums. Cell growth was measured at 30 °C for >48 h at 10 min intervals in a microplate reader (Infinite, Tecan) in three biological replicates. Growth rates were calculated as the maximum slope of the linear fit of ln(OD) over time at the exponential phase of the growth curve.

## Data processing

FastQ files from paired end sequencing of the Aβ42 library were processed using DiMSum (https://github.com/lehner-lab/DiMSum)[26], an R pipeline for analysing deep mutational scanning data. 5′ and 3′ constant regions were trimmed, allowing a maximum of 20% of mismatches relative to the reference sequence. Sequences with a Phred base quality score below 30 were discarded. At this stage, around 370 million reads passed the filtering criteria. Non-designed variants were also discarded for further analysis, as well as variants with fewer than 10 input reads in any of the replicates and variants resulting from one single nt change with fewer than 1000 input reads. Estimates from DiMSum were used to choose the filtering thresholds.

## Nucleation scores and error estimates

The DiMSum package (https://github.com/lehner-lab/DiMSum)[26] was also used to calculate nucleation scores (NS) and their error estimates for each variant in each biological replicate as:

$$\text{Nucleation score} = ES_i - ES_{WT}$$

Where $ES_i = \log(F_i\ OUTPUT) - \log(F_i\ INPUT)$ for a specific variant and $ES_{WT} = \log(F_{WT}\ OUTPUT) - \log(F_{WT}\ INPUT)$ for Aβ42 WT.

NSs for each variant were merged across biological replicates using error-weighted mean and centered to the WT Aβ42 NS. All NS and associated error estimates are available in Supplementary Data 3.

## Data analysis

**Variants in the library.** NS was obtained for 3087 unique Aβ42 variants, which were splitted into mutation classes: 751 single AA substitutions, 763 single AA insertions, 37 single AA deletions, 729 internal multi-AA deletions, 817 truncations (from one or both ends) and WT Aβ42.

In addition, nine variants (2 single AA substitutions, 6 single AA insertions, and one multi-AA deletion) were classified as non-nucleating but do not have an associated NS (i.e., they have input reads but no output reads) and are indicated as such in Fig. 2a, b and Fig. 3a. Each variant is assigned to one mutation class: deletions from position 1 or 42 are classified as truncations and not as deletions, and deletions of positions 1 and 42 are classified as single AA deletion and not as truncations. Multiple mutation classes can be combined for visualization or analysis (e.g., truncations and single deletions are included in the deletions matrix in Fig. 3a).

We assign to single AA insertions the position of the inserted AA (e.g., an insertion between positions 1 and 2 is an insertion at position 2). In the case of insertions between positions 28 and 29 (i.e., between the N and C-terminus), they are insertions at position 29 but considered N-terminal mutations.

Different mutations can result in the same coding sequence (e.g., H13Δ and H14Δ, or DAEDVGSNKGAIIGLMVGGVVIA, which is Δ2–20, Δ3–21, and Δ4–22). This is the case for single AA insertions, single and multi-AA deletions. In general, they are only considered as one coding variant but considered multiple times for visualization or if the analysis is position-specific, in figures: Figs. 2b–f, 3a, b and 5, and Supplementary Figs. 2–6 and 11a.

**Aggregation and variant effect predictors.** For the aggregation predictors (Tango, Zyggregator, Waltz, Camsol[31,34–36]), individual residue-level scores were summed to obtain a score per single AA mutation sequence. We then calculated the log value for each variant relative to the WT score. For the variant effect predictors (Polyphen and CADD[52,53]), we also calculated the log value for each single AA substitutions variant but in this case values were scaled relative to the lowest predicted score.

We also used an hydrophobicity scale[54] and a principal component from a previous study (PC1[55]) that relates strongly to changes in hydrophobicity. For each single AA substitution variant, the values of a specific AA property represent the difference between the mutant and the WT scores.

## ROC analysis

ROC curves and AUC values were built and obtained using the 'pROC' R package. The table of fAD mutations was taken from https://www.alzforum.org/mutations/app. The nucleation scores and categories for all fAD variants, as well as the criteria used to consider them as fAD, are reported in Supplementary Data 3.

## Statistics and reproducibility

Based on the transformation efficiency each variant in the designed library ($n = 3164$) is expected to be represented at least 10x at each step in selection experiments and library preparation. In terms of sequencing, reads that did not pass the QC filters using the DiMSum package were excluded (https://github.com/lehner-lab/DiMSum). The experiments were not randomized. The Investigators were not blinded to allocation during experiments and outcome assessment.

## Reporting summary

Further information on research design is available in the Nature Portfolio Reporting Summary linked to this article.

# Data availability

Raw sequencing data and the processed data table are deposited in NCBI's Gene Expression Omnibus (GEO) as GSE193837. The processed data are provided in the Supplementary Data 3. The coordinates for the PDB structures were obtained with accession: 7Q4M and 7Q4B. Source data are provided with this paper.

# Code availability

All scripts used for downstream analysis and to reproduce all figures are in https://github.com/BEBlab/DIM-abeta.

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

## Acknowledgements

M.S. was supported by a fellowship from Agencia de Gestio d'Ajuts Universitaris i de Recerca (2019FI_B 01311). Work in the lab of BB and BL is supported by the la Caixa Research Foundation project 'DeepAmyloids' (LCF/PR/HR21/52410004). Work in the lab of BB is also supported by the Spanish Ministry of Science, Innovation and Universities (PID2021-127761OB-I00 and RYC2020-028861-I, funded by MCIN/AEI/ 10.13039/501100011033, "ERDF A way of making Europe" and "ESF Investing in your future") and the CERCA Program/Generalitat de Catalunya. Work in the lab of BL is also supported by a European Research Council (ERC) Advanced Grant ('Mutanomics' 883742), the Spanish Ministry of Science, Innovation and Universities (PID2020-118723GB-I00), the Bettencourt Schueller Foundation, the AXA Research Foundation, Agencia de Gestio d'Ajuts Universitaris i de Recerca (AGAUR, 2017 SGR 1322) and the CERCA Program/Generalitat de Catalunya. We also acknowledge the support of the Spanish Ministry of Science and Innovation to the EMBL partnership and the Centro de Excelencia Severo Ochoa. We thank the Chernoff lab for providing strains and plasmids and the CRG Genomics core technology for sequencing. We also thank Andre Faure and Marta Badia for advice on data analysis, Leire Moriones for assistance with validation experiments and Xavier Salvatella and Carla Garcia for discussing our data.

## Author contributions

M.S. performed all experiments and analyses. M.S., B.L., and B.B. designed the experiments and analyses and wrote the manuscript.

## Competing interests

The authors declare no competing interests.
