## [Peer Review File · Nature Communications]

REVIEWER COMMENTS

Reviewer #1 (Remarks to the Author):

This is a very interesting study, in which the authors use a clever cell-based assay to investigate the effects of an enormous number of amino acid substitutions, insertions, deletions, and truncations on the aggregation propensity of the 42-residue amyloid-beta peptide (Ab42). From a methodological standpoint, this work is quite novel and fascinating. The results are also surprising in many ways. Although the authors find some of the well-known disease-associated mutations, they also get unexpected and counter-intuitive results, for example that increasing the hydrophobicity of Ab42 does not necessarily increase its aggregation propensity.

The authors should make major revisions to address the following points:

1. The authors define the N-terminal region to be residues 1-28 and the C-terminal region to be residues 29-42. This is not how the Ab42 sequence is normally described and does not correspond well with structural data. Typically, residues 1-11 or 1-16 are partially or fully disordered in Ab40 and Ab42 fibrils, but residues 12-21 or 17-21 form a well-ordered beta-strand-like structure. Hydrophobic residues in 17-21 are essential for the normal hydrophobic core structure. Residues 22-29 typically have an irregular structure, and then residues 30-40 or 30-42 are primarily beta-strand-like. The authors show this in Figs. 5 and Supp. Fig. 14, but their descriptions throughout the text are misleading. The authors should explain in more detail why they have chosen to define N-terminal and C-terminal regions as 1-28 and 29-42. It does not make sense to me to include residues 17-21 with residues 1-11 in the data analyses, for example.

2. A major issue is that the authors' cell-based assay involves Ab42-Sup35N fusion constructs. Consequently, they are not directly probing the properties of Ab42 mutants. Instead, they are probing the properties of Ab42-Sup35N mutants. Sup35N itself is aggregation prone, so it is not necessarily insignificant.

This needs to be stated and justified more clearly in the main text. Data in Supp. Fig. 1d suggest that Ab42-Sup35N mutants may behave similarly to Ab42 mutants in certain cases. But these data are quite limited. How can the authors be sure that the properties of Ab42-Sup35N mutants are always the same as the properties of Ab42 mutants without Sup35N?

3. Another major issue is that some intracellular Ab42-Sup35N aggregates may be cytotoxic. If a given Ab42-Sup35N construct aggregates in a cytotoxic manner, there will be no cell growth and the authors will interpret this as a low "nucleation score". It seems to me that cytotoxicity of some Ab42-Sup35N aggregates may explain some of the counter-intuitive results. For example, constructs in which Gly37 or Gly38 are replaced by hydrophobic residues generally have low nucleation scores. I would expect these constructs to be strongly aggregation prone. The authors should explain why cytotoxicity of aggregated proteins is not a problem.

4. The authors should perform their own in vitro tests of aggregation propensity (i.e., kinetics of aggregation or thermodynamic stability of the aggregates) for at least a handful of mutants for which the predictions are surprising. For example, the Gly37 or Gly38 substitutions discussed above. Or residues 29-42, which I expect will be highly aggregation-prone in vitro, but has a low nucleation score in Fig. 4a.

5. The authors say that "N-terminally truncated fragments of Abeta have been detected in AD patients". This is true, but N-terminal truncation in patients is due to proteolytic digestion of the disordered N-terminal tails AFTER aggregation, not due to enhanced aggregation propensity of N-terminally truncated monomers.

Reviewer #2 (Remarks to the Author):

This paper provides a wealth of information concerning the effects of substitutions, insertions and deletions in Abeta42. Interestingly, no single feature emerges, rather the effects are very context-dependent. This finding emphasizes the importance of performing exhaustive studies when hoping to anticipate aggregation. That biologically interesting mutants were identified to have singularly strong effects justifies the effort.

Small points:

The assay needs to be explained. Remarkably, the word "yeast" is not used even once in the main text, figure or figure legends. It only appears in supplementary materials. One gets the impression that the authors are hiding something? One familiar with DMS can infer it from Fig. 1, but the general reader might think it is done in a more (or less) relevant cell type.

Anecdotally, the authors might mention that the insertion of a hydrophobic residue between the two His residues creates a metal-binding motif that has shown to very strongly nucleate fibrillization in catalytic amyloids. <https://www.ncbi.nlm.nih.gov/pmc/articles/PMC5474797/>

Reviewer #3 (Remarks to the Author):

The article "An atlas of amyloid aggregation: the impact of substitutions, insertions, deletions and truncation on amyloid beta fibril nucleation" submitted by Senma et al. describes the analyses of numerous indel variants of the Abeta peptide and their ability and efficiency to form amyloids. The article is very clear and easy to follow. The data is presented in an efficient way. The analysis of the data is proper and discussed thoroughly regarding published results and previously established tools.

The models help to follow the major outcomes and allow a broader audience to assess the data and their interpretation.

The results from this study contribute to a better understanding of how indel variants of Abeta contribute to its fibrillization and may thus be clinically relevant.

Shortcomings:

- 1) The study relies solely on a previously established yeast assays using Sup35 fusions. I think it would be important to confirm at least some of the most interesting/surprising variants by at least one other well-established method (e.g. ThT essays in vitro) for cross-validation.
- 2) I think the implications of this study is quite clear. It would be interesting if the Discussion included an outlook how this approach can be transferred to other amyloidogenic proteins (e.g. aSyn, SOD1). This could strengthen the general impact of the manuscript.
- 3) The authors mention discrepancies between their results and results from computational analyses. I think this warrants a more detailed discussion, e.g., assessing reason why these discrepancies occur.
- 4) Abeta oligomeric species have been implicated in AD neurodegeneration. The methods and analysis tools applied here do not include this important aspect of Abeta misfolding.

AB Indels Mireia Response to Reviewers

We thank the reviewers for their enthusiasm and very helpful suggestions which we address here in a point-to-point response and with a revised version of the manuscript. The main changes that we have made are:

- Introducing a description of the primary sequence of the peptide which goes beyond the binary definition we previously employed and takes into account previous definitions of aggregation prone regions within the peptide. We have also analyzed mutational impact in these regions and updated figures accordingly (Fig 1d and Supplementary Fig. 1h).
- Measuring the cellular toxicity of a set of variants with increasing or decreasing nucleation scores (Supplementary Fig. 1f). This reveals none of these variants is toxic to yeast, so toxicity is not a confounding factor.
- Extending the comparison to in vitro kinetics to an extra 10 AB variants. The effects on nucleation upon mutation match our data in 21/25 instances (Supplementary Table 1).
- Including a comparison with rate constants extracted from an assay (HDX-MS) that does not rely on dye binding and that reports rate constants for two additional variants (Fig. 1b). Nucleation scores strongly correlate with the kinetics of secondary processes of nucleation.
- Expanding the correlation with rate constants to one extra AB variant, AB40, for which curated accurate kinetic measurement became recently available (Supplementary Fig. 1d). The significant correlation between nucleation scores and rate constants is maintained.
- Assessing performance of nucleation scores in classifying fAD mutations after including a new single amino acid substitution in AB, L17V, which was shown to cause fAD in 2021. This further improves the performance of our assay to classify fAD (Supplementary Fig. 1g, AUC=0.88).

Altogether these new data strongly support our conclusion that our deep mutagenesis data i) reports on the kinetics of aggregation and ii) reports on a mechanism that is extremely relevant to fAD.

All major changes to the text are highlighted in blue in the revised manuscript. A more in-depth point-to-point response to each reviewer's comments follows.

Reviewer #1 (Remarks to the Author):

This is a very interesting study, in which the authors use a clever cell-based assay to investigate the effects of an enormous number of amino acid substitutions, insertions, deletions, and truncations on the aggregation propensity of the 42-residue amyloid-beta peptide (Ab42). From a methodological standpoint, this work is quite novel and fascinating. The results are also surprising in many ways. Although the authors find some of the well-known disease-associated mutations, they also get unexpected and counter-intuitive results, for example that increasing the hydrophobicity of Ab42 does not necessarily increase its aggregation propensity.

The authors should make major revisions to address the following points:

1. The authors define the N-terminal region to be residues 1-28 and the C-terminal region to be residues 29-42. This is not how the Ab42 sequence is normally described and does not correspond well with structural data. Typically, residues 1-11 or 1-16 are partially or fully disordered in Ab40 and Ab42 fibrils, but residues 12-21 or 17-21 form a well-ordered beta-strand-like structure. Hydrophobic residues in 17-21 are essential for the normal hydrophobic core structure. Residues 22-29 typically have an irregular structure, and then residues 30-40 or 30-42 are primarily beta-strand-like. The authors show this in Figs. 5 and Supp. Fig. 14, but their descriptions throughout the text are misleading. The authors should explain in more detail why they have chosen to define N-terminal and C-terminal regions as 1-28 and 29-42. It does not make sense to me to include residues 17-21 with residues 1-11 in the data analyses, for example.

The distinction between N-terminus and C-terminus in our paper arises from the distinct mutational impact in the 1-28 region compared to the 29-42 region. However, we agree with the reviewer that, in our previous version of the manuscript, we jumped to this binary classification too quickly.

To avoid this, we have updated our analyses to first focus on five regions of AB that differ in the psycho-chemical properties of the primary sequence and that have previously been proposed to play different roles in the aggregation of the peptide, as well as in the final structure of AB fibrils. These include two aggregation prone regions (APR1 and APR2, AA 17-21 and 29-42 respectively), a N-terminal stretch which is disordered in many mature fibrils (AA 1-11) and two interconnecting polar stretches (AA 12-16 and 22-28). We have introduced a sentence to clearly explain the rationale behind these boundaries (page 5), as well as updating Fig. 1 and Supplementary Fig. 1 to show mutational impact in each of these regions individually. Given the clear difference in mutational impact in the regions up to residue 28, compared to the 29-42 region, for some of the further analyses in the manuscript, we have then maintained this binary classification. We have also updated Figure 5 and Supplementary Fig. 14 with a two monomer version of the AB fibrils structure, which also illustrates the different regions.

2. A major issue is that the authors' cell-based assay involves Ab42-Sup35N fusion constructs. Consequently, they are not directly probing the properties of Ab42 mutants. Instead, they are

probing the properties of Ab42-Sup35N mutants. Sup35N itself is aggregation prone, so it is not necessarily insignificant.

This needs to be stated and justified more clearly in the main text. Data in Supp. Fig. 1d suggest that Ab42-Sup35N mutants may behave similarly to Ab42 mutants in certain cases. But these data are quite limited. How can the authors be sure that the properties of Ab42-Sup35N mutants are always the same as the properties of Ab42 mutants without Sup35N?

We have previously shown that expression of the Sup35N domain alone does not result in amyloid nucleation in these strains (Seuma et al., eLife 2021). We now clearly state this in page 4. However, the question of how nucleation scores of the protein fusions correlate with in vitro aggregation rates for unfused peptide variants is overall a very important one.

We extended the correlation of the nucleation scores measured in our assay with available aggregation rate constants from the literature, with three extra variants (AB40, H6R and D7N). On one hand we could include AB40, for which curated accurate Thioflavin-T kinetic measurement became recently available (Supplementary Fig. 1d). The significant correlation between nucleation scores and rate constants is maintained. On the other, we could also include a correlation with rate constants extracted from a different type of assay (HDX-MS) (Illes-Toth, 2021), not ThT based (Fig. 1b). This dataset includes the 6 AB variants measured previously by means of ThT and two extra variants, H6R and D7N. Nucleation scores significantly correlate with nucleation also in this case.

What is more, the HDX-MS dataset (Illes-Toth, 2021), suggests that secondary nucleation is the dominant mechanism of nucleation for AB42 wild-type and all AB42 fAD variants for which an estimate of the different microscopic mechanisms of nucleation is available (Meisl, 2022). We now also show that nucleation scores from our assay best correlate with secondary processes (Fig. 1b) and have included the HDX-MS relationship between primary and secondary processes as comparison in Supplementary Fig. 1e.

We have also included an extended qualitative comparison as in Seuma et al., eLife 2021 with an extra 10 AB variants for which kinetics were measured. In these assays, mutations are scored based on whether the speed of the aggregation reaction accelerated or decelerated (Supplementary Table 1). We show that the direction in which nucleation scores increase or decrease upon mutation matches the experimental data in 21/25 instances.

Finally, two novel fAD mutations (the Uppsala mutation, Δ 19-24 and L17V) were identified in 2021, after we performed the selection experiments for this manuscript. Both of these variants have high nucleation scores and were classified by us as very likely pathogenic, providing an independent validation of our measurement. We have now included L17V, the new single amino acid substitution in AB, which further improves the performance of our assay to classify fAD (Supplementary Fig. 1g, AUC=0.88).

We mention all these points in page 4.

3. Another major issue is that some intracellular Ab42-Sup35N aggregates may be cytotoxic. If a given Ab42-Sup35N construct aggregates in a cytotoxic manner, there will be no cell growth and the authors will interpret this as a low "nucleation score". It seems to me that cytotoxicity of some Ab42-Sup35N aggregates may explain some of the counter-intuitive results. For example, constructs in which Gly37 or Gly38 are replaced by hydrophobic residues generally have low nucleation scores. I would expect these constructs to be strongly aggregation prone. The authors should explain why cytotoxicity of aggregated proteins is not a problem.

To address the reviewer's question, we have measured growth curves with and without expression of the Sup35N-AB fusion constructs (media with and without copper that induces expression from the promoter driving Sup35N-AB expression) for 9 variants with a range of nucleation scores. No Sup35N-AB variant was toxic compared to the expression of SupN alone. These data are now included as Supplementary Fig. 1f and mentioned in page 4.

4. The authors should perform their own in vitro tests of aggregation propensity (i.e., kinetics of aggregation or thermodynamic stability of the aggregates) for at least a handful of mutants for which the predictions are surprising. For example, the Gly37 or Gly38 substitutions discussed above. Or residues 29-42, which I expect will be highly aggregation-prone in vitro, but has a low nucleation score in Fig. 4a.

Our assay measures nucleation that is productive for amyloid formation. Our measured nucleation scores correlate very well with in vitro measurements of secondary nucleation kinetics and - on the basis of the agreement with human genetics - they must also report on a process that is very similar to that leading to familial AD.

It is possible - as suggested in the discussion (page 12) - that some of the non-nucleating variants undertake an alternative non-amyloid pathway of aggregation, not relevant for pathology. This could be true especially for mutations that increase the hydrophobicity of the sequence such as those mentioned by the reviewer.

A prior deep mutational scan quantified the effects of mutations on the abundance of A β fused to the enzymatic reporter DHFR, providing a set of "solubility" scores (Gray, 2019). There is no correlation between these experimentally measured changes in protein solubility and nucleation scores, suggesting that changes in solubility are not important causes of changes in nucleation in our dataset. We explicitly show this in Supplementary Fig. 9a and more explicitly state this in the revised manuscript (page 13).

A decrease in nucleation for the variants highlighted by the reviewer, substitutions to aliphatics at residues G37 and G38, is, in our opinion, very consistent with the structures of AB42 amyloid fibrils where these residues are located in a sharp turn where residues other than G are likely to be strongly disfavoured (Fig. 5a).

Finally, regarding the truncated variant 29-42, we independently validated that it has decreased nucleation compared to WT, and that a positive charge just before this sequence (i.e, K28 or R28)

can further increase nucleation, defining a minimal core for AB nucleation (page 10, Fig. 4c). It is possible that this peptide, but not 29-42 which is missing the lysine, still engages in a salt bridge with the carboxy-terminus of another monomer, as observed for one of the two polymorphs of fibrils extracted from AD brains.

5. The authors say that "N-terminally truncated fragments of Abeta have been detected in AD patients". This is true, but N-terminal truncation in patients is due to proteolytic digestion of the disordered N-terminal tails AFTER aggregation, not due to enhanced aggregation propensity of N-terminally truncated monomers.

While several truncated or modified versions of AB have been detected in AD brains, that the action of exopeptidase occurs exclusively after fibril deposition is still highly debated (Dunys, 2018 and Valverde, 2021). Modified and truncated versions of AB are also the target of several therapeutic strategies aimed at inhibiting AB aggregation and toxicity. With this perspective in mind we think it's important to point out the increased nucleation propensity of many truncated variants in our dataset.

Reviewer #2 (Remarks to the Author):

This paper provides a wealth of information concerning the effects of substitutions, insertions and deletions in Abeta42. Interestingly, no single feature emerges, rather the effects are very context-dependent. This finding emphasizes the importance of performing exhaustive studies when hoping to anticipate aggregation. That biologically interesting mutants were identified to have singularly strong effects justifies the effort.

Small points:

The assay needs to be explained. Remarkably, the word "yeast" is not used even once in the main text, figure or figure legends. It only appears in supplementary materials. One gets the impression that the authors are hiding something? One familiar with DMS can infer it from Fig. 1, but the general reader might think it is done in a more (or less) relevant cell type.

We thank the reviewer for pointing this out and have stated clearly that yeast is the model employed for this massively parallel nucleation assay (page 2).

Anecdotally, the authors might mention that the insertion of a hydrophobic residue between the two His residues creates a metal-binding motif that has shown to very strongly nucleate fibrillization in catalytic amyloids. <https://www.ncbi.nlm.nih.gov/pmc/articles/PMC5474797/>

This is a very interesting observation which we have now included in the discussion of our results (page 7).

Reviewer #3 (Remarks to the Author):

The article “An atlas of amyloid aggregation: the impact of substitutions, insertions, deletions and truncation on amyloid beta fibril nucleation” submitted by Senma et al. describes the analyses of numerous indel variants of the A β peptide and their ability and efficiency to form amyloids.

The article is very clear and easy to follow. The data is presented in an efficient way. The analysis of the data is proper and discussed thoroughly regarding published results and previously established tools.

The models help to follow the major outcomes and allow a broader audience to assess the data and their interpretation.

The results from this study contribute to a better understanding of how indel variants of A β contribute to its fibrillization and may thus be clinically relevant.

Shortcomings:

1) The study relies solely on a previously established yeast assays using Sup35 fusions. I think it would be important to confirm at least some of the most interesting/surprising variants by at least one other well-established method (e.g. ThT essays in vitro) for cross-validation.

We agree that the question of how nucleation scores of the protein Sup35 fusions correlate with in vitro aggregation rates for unfused peptide variants is overall a very important one.

In our previous work we showed that nucleation scores strongly correlated with kinetic rate constant for primary and secondary nucleation from in vitro ThT measurements for a set of variants available at that time. Here, we extended the correlation of the nucleation scores measured in our assay with available aggregation rate constants from the literature, with three extra variants (AB40, H6R and D7N).

On one hand we could include AB40, for which curated accurate Thioflavin-T kinetic measurement became recently available (Supplementary Fig. 1d). The significant correlation between nucleation scores and rate constants is maintained. On the other, we could also include a correlation with rate constants extracted from a different type of assay (HDX-MS) (Illes-Toth, 2021), not ThT based (Fig. 1b). This dataset includes the 6 AB variants measured previously by means of ThT and two extra variants, H6R and D7N. Nucleation scores significantly correlate with nucleation also in this case.

What is more, the HDX-MS dataset (Illes-Toth, 2021), suggests that secondary nucleation is the dominant mechanism of nucleation for AB42 wild-type and all AB42 fAD variants for which an estimate of the different microscopic mechanisms of nucleation is available (Meisl, 2022). We now also show that nucleation scores from our assay best correlate with secondary processes (Fig. 1b) and have included the HDX-MS relationship between primary and secondary processes as comparison in Supplementary Fig. 1e.

We have also included an extended qualitative comparison as in Seuma et al., eLife 2021 with an extra 10 AB variants for which kinetics were measured. In these assays, mutations are scored based on whether the speed of the aggregation reaction accelerated or decelerated

(Supplementary Table 1). We show that the direction in which nucleation scores increase or decrease upon mutation matches the experimental data in 21/25 instances.

Finally, two novel fAD mutations (the Uppsala mutation, Δ 19-24 and L17V) were identified in 2021, after we performed the selection experiments for this manuscript. Both of these variants have high nucleation scores and were classified by us as very likely pathogenic, providing an independent validation of our measurement.

Altogether these new data strongly contribute to the validation of this assay as an assay that i) reports on kinetics of aggregation and ii) reports on a mechanism that is extremely relevant to fAD.

We mention all these points in page 4.

2) I think the implications of this study is quite clear. It would be interesting if the Discussion included an outlook how this approach can be transferred to other amyloidogenic proteins (e.g. aSyn, SOD1). This could strengthen the general impact of the manuscript.

We have added a concluding sentence which stresses that the impact of this approach goes beyond understanding A β 42 nucleation, but rather opens up the path to investigate different primary sequences that are able to nucleate amyloids. When it comes to human amyloids, this will also allow us to identify those proteins whose mutations lead to disease as a result of increased nucleation, vs those where aggregation is irrelevant to disease.

3) The authors mention discrepancies between their results and results from computational analyses. I think this warrants a more detailed discussion, e.g., assessing reason why these discrepancies occur.

We suspect that the relatively poor performance of most aggregation predictors is due to small size of the experimental datasets used to train them as well as the bias of these datasets towards short peptides and/or very specific protein sequences. For example, Waltz was trained on 287 hexapeptides and tested experimentally on 30 peptides derived from Uniprot. The accuracy of Tango predictions was evaluated using 179 peptides (from 21 different proteins) whose aggregation was previously characterized in the literature and tested experimentally on 71 peptides derived from only 3 proteins (prion protein, lysozyme, beta-2-microglobulin). In terms of predictions of mutational impact, Tango performance was assessed for just 4 fAD variants of A β 42. Finally, Zyggregator was tested on 79 proteins for which aggregation rates were available in the literature and 59/79 are variants of the same protein, Acylphosphatase. We have added a sentence to the manuscript discussing this on page 7.

4) Abeta oligomeric species have been implicated in AD neurodegeneration. The methods and analysis tools applied here do not include this important aspect of Abeta misfolding.

We now mention this in the revised version of the manuscript on page 2. Nucleation scores correlate very well with in vitro measurements of secondary nucleation kinetics and - on the basis of the agreement with human genetics - they must also report on a process that is very similar to that leading to familial AD. Since secondary nucleation is the recognised driving mechanism in the generation of toxic AB42 oligomers (Cohen, 2013) it is reasonable to think that our measurements are reporting on oligomer formation, but we prefer not to speculate on something that is not directly measured by us.

REVIEWERS' COMMENTS

Reviewer #1 (Remarks to the Author):

In this revised manuscript, the authors have made major revisions that adequately address my comments about the original manuscript. Revisions include clarifications in the text, new analyses, and new experimental results. No further revisions are needed.

Reviewer #3 (Remarks to the Author):

The authors revised the manuscript in a highly efficient way and added new data and discussions points, which make this article much stronger and highly relevant to the field. The authors also thoroughly addressed all issues brought forward by the three reviewers. I think the article is ready for publication now.